# Surface-Enhanced Raman Scattering (SERS) Substrates Based on Ag-Nanoparticles and Ag-Nanoparticles/Poly (methyl methacrylate) Composites

**DOI:** 10.3390/polym15122624

**Published:** 2023-06-09

**Authors:** Mayra Matamoros-Ambrocio, Enrique Sánchez-Mora, Estela Gómez-Barojas

**Affiliations:** 1Centro de Investigaciones en Dispositivos Semiconductores (CIDS-ICUAP), Benemérita Universidad Autónoma de Puebla, P.O. Box 196, Puebla 72570, Mexico; mayra.matamoros@alumno.buap.mx (M.M.-A.); egomez@ifuap.buap.mx (E.G.-B.); 2Institute of Physics, Benemérita Universidad Autónoma de Puebla, Eco Campus Valsequillo, Independencia O 2 sur No. 50, San Pedro Zacachimalpa, P.O. Box J-48, Puebla 72960, Mexico

**Keywords:** Ag nanoparticles, PMMA microspheres, SERS substrates

## Abstract

SERS substrates formed by spherical silver nanoparticles (Ag-NPs) with a 15 nm average diameter adsorbed on Si substrate at three different concentrations and Ag/PMMA composites formed by an opal of PMMA microspheres of 298 nm average diameter were synthesized. The Ag-NPs were varied at three different concentrations. We have observed from SEM micrographs, in the Ag/PMMA composites, the periodicity of the PMMA opals is slightly altered as the Ag-NP concentration is increased; as a consequence of this effect, the PBGs maxima shift toward longer wavelengths, decrease in intensity, and broaden as the Ag-NP concentration is increased in the composites. The performance of single Ag-NP and Ag/PMMA composites as SERS substrates was determined using methylene blue (MB) as a probe molecule with concentrations in the range of 0.5 µM to 2.5 µM. We found that in both single Ag-NP and Ag/PMMA composites as SERS substrates, the enhancement factor (EF) increases as the Ag-NP concentration is increased. We highlight that the SERS substrate with the highest concentration of Ag-NPs has the highest EF due to the formation of metallic clusters on the surface, which generates more “hot spots”. The comparison of the EFs of the single Ag-NP with those of Ag/PMMA composite SERS substrates shows that the EFs of the former are nearly 10-fold higher than those of Ag/PMMA composites. This result is obtained probably due to the porosity of the PMMA microspheres that decreases the local electric field strength. Furthermore, PMMA exerts a shielding effect that affects the optical efficiency of Ag-NPs. Moreover, the metal–dielectric surface interaction contributes to the decrease in the EF. Other aspect to consider in our results is in relation to the difference in the EF of the Ag/PMMA composite and Ag-NP SERS substrates and is due to the existing mismatch between the frequency range of the PMMA opal stop band and the LSPR frequency range of the Ag metal nanoparticles adsorbed on the PMMA opal host matrix.

## 1. Introduction

Since the discovery of the Raman effect in 1928 by Sir C. V. Raman and K.S. Krishan [1], Raman spectroscopy has been a technique widely used in many fields of research, such as chemistry, physics, biology, and materials science [2,3]. This technique is based on the inelastic scattering effect and provides detailed information about the vibrational modes of a system under analysis. The main problem for this technique is that the inelastic scattering signal is very weak, approximately one in 10^6^–10^10^ photons scatters inelastically [4]; as a result, its application in detecting organic compounds with concentrations as low as parts per million (ppm, mg/L) is limited.

The surface-enhanced Raman scattering (SERS) effect overcomes this limitation as it considerably increases weak Raman signals by many orders of magnitude. This enhancement is mainly caused by the local electric field induced near the metallic nanoparticle (NP) surface [5]. When incident light frequency is in resonance with the oscillations of electrons in a metallic nanoparticle, all electrons will be driven to oscillate collectively in an optical phenomenon known as localized surface plasmon resonance (LSPR) [6]. This results in a strong light scattering by the metallic NPs producing an enhancement of the local electric field in the “hot spots” [7]. Therefore, the molecules in these hot spots significantly enhance their Raman signal intensities, improving the sensitivity and detection and, in some cases, single-molecule detection is possible [8].

The SERS effect was reported for the first time in 1974 by Fleischmann et al. [9] who observed an enhanced Raman scattering signal using pyridine adsorbed on a rough silver electrode. Subsequently, metal NPs of many sizes and shapes have been fabricated as SERS substrates. In this context, Ag-NPs are frequently applied in optical field detection because of their better optical properties. Furthermore, the Ag-NPs are more effective than Au-NPs due to their ability to manifest a much more powerful surface electric field undergoing a plasmonic excitation state [10,11]. Therefore, Ag-NPs exhibit an EF higher than that of Au-NPs [12]. For this reason, there has been great interest in developing different synthesis methods for Ag-NP SERS substrates, and among them is chemical reduction [13,14] where a metal precursor, such as silver nitrate, is chemically reduced by a reductant, such as sodium borohydride or sodium citrate, to generate Ag-NPs.

Nowadays, the composites formed with polymers and metallic NPs have attracted great interest due to their multifunctionality and are potential alternatives for developing efficient and scalable SERS substrates [15]. A polymer can be selected to provide either a stimuli-responsive platform or a porous polymeric matrix to facilitate the diffusion and entrapment of biomolecules under analysis [15,16,17]. For example, Gushiken et al. [18] have described the preparation of a nanocomposite formed by gold ion-implanted in PMMA. The Au-NPs were synthesized ensuring that the spacing between them is less than 10 nm, and they were located below the PMMA surface. Taking advantage of the swelling effect of PMMA, the analyte molecules were guided to the hot spot regions, and the SERS signals were intensified.

The polymeric matrix was used to fabricate flexible SERS substrates. Wang et al. [19] reported the preparation of Ag-NP/PMMA flexible SERS substrate that offers good transparency and flexibility to detect residual pesticides in situ and real-time on curved fruit surfaces. Tang et al. [20] have reported a SERS substrate based on hydrophilic–hydrophobic Ag-modified PMMA for a clinical application, which can unexpectedly inhibit the ‘‘coffee-ring phenomenon”. The target analytes were consequently enriched in the SERS-active Ag regions using hydrophobic PMMA. Recently, two different polymers have been combined to generate highly ordered and controlled three-dimensional (3D) structures. Chang et al. [21] have developed novel SERS substrates formed by a PMMA film in which polystyrene (PS) nanospheres were assembled. Then, the PS nanospheres were selectively removed using cyclohexane, leaving a PMMA film with a volcano-shaped structure, and then, the PMMA film was coated with Au-NPs. The intensity of Raman signals using this volcano-shaped structure was much higher than those obtained from the pristine PMMA films and PMMA films with adsorbed Au-NPs.

In the literature, several researchers [16,18,19,22] have studied SERS substrates formed by metal/polymer composites. These works report the improvement of the Raman signal intensity of the analyte and the EF. However, a comparison of the results of single Ag-NP SERS substrates and metal/polymer composite SERS substrates is not reported, and there are no reports explaining the possible influence that a polymer could exert on metallic nanoparticles that can enhance the Raman bands of the analyte.

In our previous study, we reported the synthesis of PMMA polymer microspheres using a surfactant-free emulsion polymerization method in which, by varying the monomer and initiator, we could control the diameter of PMMA microspheres [23]. Therefore, the advantages of a polymeric composite and PMMA are properties that stand out, such as good mechanical stability, biocompatibility, high optical transmission in the visible region, and low autofluorescence [24]. 

In this study, we have performed a systematic analysis to establish the influence of the Ag-NP concentration on SERS substrates, to determine the effect of PMMA microspheres used as a host matrix on both Ag-NPs forming composites used as SERS substrates, and to determine the minimum MB concentration detectable using both SERS substrates systems.

## 2. Materials and Methods

### 2.1. Materials

Methyl methacrylate (MMA; C_5_H_8_O_2_; 99%), 2,2′-azobis (2-methylpropionamidine) dihydrochloride (V50; C_8_H_20_Cl_2_N_6_; 97%), silver nitrate (AgNO_3_; 99%), sodium borohydride (NaBH_4_; 98%), bis(p-sulfonatophenyl)phenylphosphine dihydrate dipotassium salt (BSPP; C_18_H_13_K_2_O_6_PS_2_·2H_2_O; 97%), and methylene blue (MB; C_16_H_18_ClN_3_S; 97%) were purchased from Sigma-Aldrich (St. Louis, MO, USA). Sodium citrate dihydrate (Na_3_Ct; C_6_H_5_Na_3_O_7_·2H_2_O; 99%), sulfuric acid (H_2_SO_4_; 98%), and hydrogen peroxide (H_2_O_2_; 32%) were purchased from J. T. Baker (Phillipsburg, NJ, USA). All chemicals were used without further purification. Deionized water (DI; 18.2 MΩ-cm) was used in all experiments. n-type (100) Si wafers (Wafer World, Inc., West Palm Beach, FL, USA) were used as supports for SERS substrates in all experiments.

### 2.2. Synthesis of PMMA Microspheres

The details of the synthesis of PMMA microspheres have been described in a previous report [23]. Briefly, in a system like the one shown in Figure 1, 160 mL of DI water and 285 mmol of methyl methacrylate were mixed and kept under vigorous magnetic stirring (~400 rpm) and constant N_2_ bubbling (0.2 cm^3^ s^−1^) flux. The temperature was raised to 75 °C, and 1.65 mmol of initiator V-50 was added. Since the polymerization is an exothermic reaction, the systemic temperature reached 95 °C and was kept constant for 2 h. Finally, the reaction was allowed to cool down to room temperature; then, the rinsing procedure of PMMA microspheres was carried out using a centrifuge HERMLE Z36 HK (Hermle Labortechnik GmbH, Wehingen, Germany) at 6000 rpm for 30 min. This process was repeated three times. Then, the supernatant was removed using decantation, and the solid was redispersed in 50 mL of DI water to get a 1.2 mg/L concentration.

### 2.3. Synthesis of Ag-Nanoparticles

Ag-NPs were synthesized by chemical reduction of Ag ions [25,26]. For this purpose, in a 250 mL three-necks boiling flask, 95 mL of DI water, 0.5 mL of silver nitrate solution (20 mM), and 1 mL of sodium citrate solution (30 mM) were combined. The mixture was kept under vigorous magnetic stirring in an ice bath (~5 °C) and constant N_2_ bubbling (0.2 cm^3^ s^−1^) flux for 1 h. Then, 1 mL of NaBH_4_ solution (50 mM) was rapidly added to the reaction mixture. After 15 min, 100 µL of NaBH_4_ solution was added dropwise, and this process was repeated every 2 min until 500 μL of NaBH_4_ solution was added. At this moment, 1 mL of BSPP solution (5 mM) was mixed with 1 mL of NaBH_4_ solution and added dropwise to the reaction system. In an ice bath, the resulting Ag-NP colloid was kept under magnetic stirring for 16 h. Then, the colloid was rinsed several times using centrifugation at 20,000 rpm with DI water. Finally, Ag-NPs were redispersed in 50 mL of DI water, and a 1.88 mM solution was obtained.

### 2.4. Preparation of SERS Substrates

A set of Si wafers of 1 cm^2^ area and 0.5 mm thickness were cleaned with a piranha solution (H_2_SO_4_:H_2_O_2_ 7:1 *v*/*v*) for 20 min in an ultrasonic bath, rinsed with DI water, and blow dried with N_2_. Then, 100 µL of Ag-NPs (0.94 mM) was deposited directly on the polished surface of a Si substrate, and the solvent was removed using evaporation at 70 °C in a tubular oven (Thermo Scientific Lindberg/Blue M; TF55030A-1; Waltham, MA, USA) under constant N_2_ gas flow (0.2 cm^3^ s^−1^) for 1 h; this substrate constitutes our SERS substrate. In the same way, two more SERS substrates were prepared but deposited on one Ag-NP at 1.41 mM and the other at 1.88 mM. These SERS substrates were labeled Ag-S1, Ag-S2, and Ag-S3 to shorten the notation.

### 2.5. Preparation of Ag/PMMA Composites

Three Ag/PMMA composite film samples were prepared using the co-deposition method on Si wafers. First, 270 µL of PMMA microspheres (1.2 mg/L) was dispersed in 40 mL of DI water under vigorous magnetic stirring to get an 8 mg/mL concentration. Afterward, in 10 mL beakers, 5 mL of this PMMA solution and 1 mL of Ag-NP solution were mixed, and each had one of the concentrations: 0.94 mM, 1.41 mM, and 1.88 mM for 30 min, respectively. Then, a clean Si substrate was introduced vertically in each solution to ensure the formation of the film; the solvent was slowly removed using evaporation at 75 °C in an oven (Thermolyne FB1418M, Thermo Fisher Scientific; Waltham, MA, USA) for 24 h. The resulting Ag/PMMA composites were labeled as Ag/PMMA-S1, Ag/PMMA-S2, and Ag/PMMA-S3.

### 2.6. Materials Characterization

The morphology and elemental composition were performed using a scanning electron microscope (SEM; JSM-7800F, JEOL, Tokyo, Japan) equipped with an energy-dispersive X-ray spectrometer (EDS; Oxford Instruments, X-Max; Concord, MD, USA). The Raman spectra were recorded for the samples using micro-Raman spectroscopy (μ-RS; LabRAM HR-Olympus, Horiba Jobin Yvon Inc., Edison, NJ, USA), in a backscattering geometry using the 632.8 nm emission line of a He–Ne laser as the excitation source with 9.26 mW of power. Optical characterization was performed using a spectrometer UV-Vis-NIR Varian Cary 5000 (Agilent Technologies Inc., Santa Clara, CA, USA) equipped with a DRA-CA-30I (Agilent Technologies Inc., Santa Clara, CA, USA) accessory and a Labsphere (Agilent Technologies Inc., Santa Clara, CA, USA) as a reference to record diffuse reflectance spectra (DRS). Additionally, the system is equipped with a variable angle specular reflectance accessory (VASRA) and an Al mirror as a reference. The hydrodynamic diameters of Ag-NPs were measured in dilute aqueous solutions with a dynamic light scattering (DLS) Malvern Zetasizer Nano ZS (Malvern Panalytical Ltd., Malvern, Worcestershire, UK) system with backscattering geometry, and a 4.0 mW power He–Ne laser is coupled to the system and an emission line λ = 632.8 nm is used as the excitation wavelength. The zeta potential measurements were also carried using this system. 

The physical nitrogen adsorption–desorption isotherms were obtained using the nitrogen physisorption technique at the boiling temperature of liquid nitrogen (−196 °C), using an AS1-C-MS Quantachrome Physisorption Analyzer (Boynton Beach, FL, USA). Previously, the solid sample was degassed under vacuum conditions (~10^−3^ torr) at 150 °C for 24 h. The Brunauer–Emmett–Teller (BET) specific surface area was quantified using the adsorption–desorption data in the relative pressure range of 0.01 to 0.30. The total pore volume was estimated from the adsorption data at 0.99 relative pressure. The pore size distribution was obtained applying the standard Barrett–Joyner–Halenda (BJH) method to the adsorption data.

### 2.7. SERS Measurements

This section describes the procedure to test the performance of the Ag-NP and Ag/PMMA composite samples as SERS substrates; for this purpose, MB was used as a probe molecule. MB solution (50 µL) of different concentrations (0.5, 1.0, 1.5, 2.0, and 2.5 × 10^−6^ M) was dropped on SERS substrates and dried in a tubular oven at 50 °C using a constant N_2_ gas flow (0.2 cm^3^ s^−1^) for 30 min. Then, the Raman spectra were recorded for SERS substrates using a micro-Raman spectrometer system with an ×50 objective lens and D 0.3, D 0.6, D 1, D 2, and D 3 filters to obtain signal attenuation factors of 1/2, 1/4, 1/10, 1/100, and 1/1000, respectively. The exposure time was 5 s, and the accumulation time was 10 s. The laser beam spot diameter on the sample was about 4 µm. Raman spectra were observed in three different regions of each SERS substrate in the radial direction, and the average values are reported here.

The same MB solutions were deposited on Si wafers under the same conditions, and normal Raman spectra were acquired as reference.

## 3. Results and Discussion

### 3.1. Characterization of SERS Substrates

Figure 2a–c display SEM micrographs, with 200 nm scale bars, of Ag-NPs deposited on silicon substrates. It is seen that the Ag-NPs have quasi-spherical shapes with an average diameter of 15 ± 3.7 nm, according to the size distribution histogram shown in the inset (Figure 2a). Figure 2b shows Ag-NPs with different sizes and some small agglomerates. In Figure 2c, as expected, the Ag-S3 SERS sample presents larger Ag-NP agglomerates because the drop-casting of Ag-NPs was performed with the Ag-NP solution with the highest concentration.

We would like to point out that, in Ag-NP agglomerates, the Ag-NPs are close-spaced (2–4 nm [27]), and these metallic clusters (hot spots) as well as the cavities of rough surfaces favor the local enhancement of the electromagnetic field associated with the incident optical field.

In Figure 2d, the UV–Vis absorption spectrum of each constituent of the Ag-NP synthesis solution is shown. In the absorbance spectrum of the Ag-NP solution, there is a clear band at 395 nm corresponding to the surface plasmon that represents a typical dipole resonance of spherical nanoparticles. However, the two bands located at 240 and 280 nm correspond to the electronic transitions, n → π* and π → π*, associated only with the BSPP and not with NaBH_4_ or Na_3_Ct. DLS measurements determined that the colloidal solution mainly contains Ag-NPs with hydrodynamic diameters (D_h_) of 20 ± 5.2 nm. The difference between the values of the diameters is determined from SEM micrographs, and DLS is the phase state of the sample. SEM micrographs were performed on solid samples, while DLS measurements were performed on colloidal solutions, that is, where oxygen atoms are attached to the BSPP sulfonate groups that are anchored to the surface of Ag-NPs. These DLS results are supported by the zeta potential (ζ) measurements that give a value of −42.5 mV, confirming the presence of BSPP on the surface of Ag-NPs.

Figure 3a shows SEM micrographs of PMMA microsphere sample. It is observed that all PMMA particles have a spherical shape, and this result confirms the spherical term used throughout the text. According to the size distribution histogram appearing as an inset, the average diameter of PMMA microspheres is 300 ± 8 nm. Figure 3b–d show SEM images of Ag/PMMA composites with different nominal Ag-NP concentrations. As insets, SEM images at ×100,000 have been included for better visualization. From these magnified micrographs, we can infer that the amount of Ag-NPs (white points) adsorbed on the PMMA microsphere surface increases as the Ag-NP solution concentration increases.

Furthermore, it is observed that the spherical shape of PMMA microspheres is deformed due to the presence of Ag-NPs (see the insets in Figure 3c,d). As previously discussed, the negative ζ potential that the Ag-NPs present is associated with the sulfonate group of BSPP, while the PMMA microspheres have a positive ζ potential due to the protonation of the carboxyl group [23]. Therefore, when the composite is formed, a hydrogen bridge type of interaction forms on the surface between the sulfonate and the hydrogen of the protonated carbonyl group of the Ag-NPs and PMMA microspheres, respectively, as shown in Figure 4. This interaction causes a deformation of the spherical shape of PMMA microspheres and, consequently, of the self-assembly order of the Ag-NPs. It seems that the degree of deformation depends on the amount of Ag-NPs.

Figure 5 shows overlapped Raman spectra of Ag/PMMA composites in the range from 150 to 2000 cm^−1^; the Raman spectra of single PMMA and pristine Si substrate have been included for comparison. The assignment of the bands observed in the Raman spectra is presented in Table 1 [23]. Generally, when Ag-NPs are adsorbed on the PMMA surface, the Raman bands of single PMMA shift towards shorter wavenumbers (~5 cm^−1^). This fact is probably due to the interaction between Ag-NPs and PMMA microspheres (See Table 1). However, the Raman bands of the composites are the same as those of pristine PMMA; no additional bands are observed associated with Ag-NPs.

On the other side, Figure 6 shows diffuse reflectance spectra (expressed as absorbance) of PMMA microspheres and Ag/PMMA composites with different amounts of Ag-NPs. In all spectra, the bands in the NIR range appear at the same wavelength positions as those of PMMA microspheres, but their amplitudes diminish as the amount of Ag-NPs in the composite increases. In the visible range of the absorbance spectra of the Ag/PMMA composites (Figure 6 inset), there is a small band with maximum at 416, 413, and 412 nm for the Ag/PMMA-S1, Ag/PMMA-S2, and Ag/PMMA-S3 samples, respectively. This band is associated with an LSPR of Ag-NPs [28]. The intensity of the band increases slightly as with increase in the Ag-NP concentration. 

In the UV region, the absorption edge appears at approximately 270 nm and is associated with an indirect electronic transition in the polymer [23], corresponding to an energy band gap of 4.58 eV. When the Ag-NPs are incorporated to form SERS substrate composites, a slight shift of the cut wavelengths towards shorter wavelengths is observed corresponding to an Eg shift (from 4.58 to 4.68 eV) as the amount of Ag-NPs adsorbed on PMMA microsphere surface increases. This energy shift and the displacement of the LSPR band are associated with the surface interaction between metal and dielectric surface.

Figure 7a–d show specular reflectance spectra for 20–70° incident angles with 10° increment. For the PMMA opals (Figure 7a), at 20°, a well-defined photonic band at approximately 800 nm is observed. This band diminished in intensity and shifted towards shorter wavelengths as the incident angle was varied from 20 to 70°, which is a characteristic of ordered systems. When a small amount of Ag-NPs (Ag/PMMA-S1) is incorporated into the composite, the photonic band diminishes in amplitude, its shape widens, and its peak position shifts towards shorter wavelengths (Figure 7b). The same behavior is observed in the Ag/PMMA-S2 (Figure 7c) and Ag/PMMA-S3 (Figure 7d) substrates. These effects are better visualized in Figure 7e where the specular reflectance spectra at 20° of all systems under analysis are shown. It is observed that the maximum photonic band position shifts towards larger wavelengths, and the amplitude diminishes as the amount of Ag-NPs in the composites is increased. Finally, in Figure 7f, a plot of (λ^2^) versus (sin^2^ θ) is shown where the linear relationship probes whether the Bragg–Snell condition is satisfied [29]. 

Figure 8 presents graphs of the adsorption–desorption isotherms and the pore size distribution of PMMA microspheres. The PMMA microspheres show a Type IV isotherm with a H1 hysteresis loop (Figure 8a) [30], with a pronounced capillary condensation step above 0.8 relative pressure, suggesting a well-developed porosity material. The specific surface area is 95 m^2^/g and the pore size distribution shows a maximum of 158 Å (determined using the BJH method), with a pore average diameter of 109 Å (Figure 8b). The type of isotherm and the hysteresis loop are in accord with those reported by B. Yu et al. [31]. They report the synthesis of PMMA microspheres of 250 nm diameter with toluene and dibutylphthalate (DBP). However, the average pore size is approximately 24.4 nm (244 Å); however, they do not report the specific surface area of PMMA microspheres. 

### 3.2. SERS Performance

#### 3.2.1. Ag-NP SERS Substrates

In Figure 9b–d, the SERS effect is clearly seen when Ag-NPs are present in the standard solution as the Raman bands intensities increased by four orders of magnitude with respect to the standard solutions (Figure 9a). Although all bands present significant increments in their intensities, the bands located at 455 cm^−1^ (assigned to the backbone deformation of the C-N-C bonds) and at 1632 cm^−1^ (assigned to the stretching of the C-C ring bond) [32] are the ones with higher intensities. However, when using Ag-NPs as SERS substrates, a slight shift (~6 cm^−1^) in all MB bands towards larger wavelengths is observed. 

These changes are probably due to the interaction of the MB molecules with the SERS substrate, which affects the vibrational modes of the molecule [33]. Nevertheless, this shift is very small. It should be noted that no new band indicates a modification of the MB analyte due to the presence of Ag-NPs. Furthermore, we can observe that the bands with highest intensities originate from the Raman spectra of the Ag-S3 substrate where the intensities increases by four orders of magnitude compared to those of the standard MB solutions. Meanwhile, for Ag-S2 substrate, the bands intensities increase by three orders of magnitude. Finally, the Ag-S1 substrate presented an increment of two orders of magnitude. This result shows that the SERS substrates under analysis have a high detection sensitivity of MB concentrations, i.e., as low as 5.0 × 10^−10^ M, using the Ag-S3 SERS substrate.

To evaluate the efficiency of a SERS substrate, we have calculated the EF by choosing the band located at 1632 cm^−1^ of the Raman spectra shown in Figure 9 as the band of interest and using Equation (1) [34]:(1)$$\mathrm{EF}=\frac{I_{\mathrm{SERS}}/N_{\mathrm{SERS}}}{I_{\mathrm{Raman}}/N_{\mathrm{Raman}}}$$
where I_SERS_ and I_Raman_ denote the intensities of the characteristic band located at 1632 cm^−1^ measured from Raman spectra of MB adsorbed on the SERS substrates and the non-SERS substrate (MB/Si). N_SERS_ and N_Raman_ are the numbers of excited molecules in the SERS substrate and the non-SERS substrate, representing the number of scatters in the samples. N_Raman_ depends on the analyte concentration and the specific area of measurement. On the other side, N_SERS_ is related to the number of molecules of the analyte adsorbed on the SERS surface substrate.

Thus, N_Raman_ was calculated considering MB molecules contained in 50 μL of any used concentration. When methylene blue is dropped on the SERS substrate, it spreads, forming a circular area of approximately 1.0 cm diameter or an area of 7.85 × 10^7^ μm^2^. This area is much larger than the area generated by focusing the laser with ×50 objective lens of the optical microscope on the sample, which is about 12.57 μm^2^. The latter becomes the area of interest used in the calculations. Therefore, N_Raman_ is the number of MB excited molecules during the Raman spectra acquisition. It is worth mentioning that the MB distribution on the substrate is not homogeneous; there are empty regions. Therefore, the relationship between the whole area covered by the MB dropped on the substrate, the MB molecular area, and the MB molecules existing on the area of interest were determined.

To calculate N_Raman_ and N_SERS_, it was asssumed that the MB molecules were adsorbed on a silicon substrate and on the Ag-NPs with a horizontal orientation, that is, with the nitrogen atom located in the central aromatic ring [35]. In this case, the area of the flat base of a MB molecule is 4.56 × 10^−7^ μm^2^. Now, the ratio of the whole area of the MB drop and the MB molecular area suggests 1.72 × 10^14^ molecules of MB are required to cover the whole drop area. However, the highest MB concentration used (2.5 × 10^−6^ M) contains only 7.5 × 10^13^ molecules and makes us infer that the whole drop area is not occupied by MB molecules, as mentioned before. Then, the occupied area percentage for each concentration were calculated. On the other side, to calculate N_SERS_, it is necessary to consider that only MB molecules on Ag-NPs be responsible for the SERS intensity. Therefore, a detailed analysis of the SEM micrographs of SERS substrates was performed using the ImageJ64 software, which allowed us to determine the approximate number of Ag-NPs present inside the incident laser beam area. For example, for the Ag-S1 substrate, it was found that the laser beam area is about 10% of the area occupied by Ag-NPs. Once again, taking into account the occupied area by a MB molecule adsorbed horizontally and considering that all MB molecules were deposited on Ag-NPs, there are a total of 1.35 × 10^3^ MB molecules present for the SERS effect. Finally, the EF values were calculated and listed in Table 2 for the remaining SERS substrates: Ag-S2 and Ag-S3.

Figure 10 shows a plot of EF data of each substrate: Ag-S1, Ag-S2, and Ag-S3 as a function of the MB concentration. First, for a fixed MB concentration, the EF increases as the Ag-NP concentration on the SERS substrates increases. This fact could be due to the local enhancement of the incident optical field by protrusions or in cavities of rough surfaces either of single Ag-NP or clusters. Besides, for each Ag-NP sample used as a SERS substrate, the EF increases linearly as the MB concentration increases. This effect could be due to a chemical interaction between the MB molecules and the Ag-NPs. Thus, we could infer that the EF here observed has both a physical and chemical contributions. Furthermore, it is inferred from Figure 10 that when the EF increases linearly for each SERS substrate as the MB concentration increases, the slopes are not the same, and the one for the Ag-S3 SERS substrate is the highest. The SERS Ag-S3 substrate presents the best efficiency produced by the highest Ag-NP concentration, which implies the highest number of hot spots responsible for enhancing the electromagnetic field intensity.

It is observed that the EF of the MB Raman band of the Ag-S3 substrate (1.88 mM) presents a linear behavior with the greatest slope compared to Ag-S2 (1.41 mM) and Ag-S1 (0.94 mM) SERS substrates, even when the difference in concentrations among them is n∆, where ∆ = 0.47 mM and n = 2, 3, and 4. 

This result is due to the highest formation of Ag-NP clusters in the Ag-S3 SERS substrate in comparison to the other two substrates, as seen in the SEM micrographs (Figure 2b,c). This fact supports the existence of more hot spots that increase the local electromagnetic field and, consequently, enhance the Raman scattering band intensities [36].

#### 3.2.2. Ag/PMMA Composite SERS Substrates

In the same way as Ag-NPs substrates, the Ag/PMMA composites were analyzed as SERS substrates using MB as a probe molecule. Figure 11 shows the Raman spectra of MB adsorbed on Ag-NPs/PMMA composites used as SERS substrates. 

We highlight that in these spectra no band associated with PMMA are observed. This result suggests that the PMMA substrate does not interfere with the Raman spectra of the SERS substrate, and no shift is observed in the MB bands that could indicate a modification of PMMA or Ag-NP presence.

In the same way as in the results for Ag-NPs as SERS substrates, the Raman intensity increases linearly as the MB concentration increases. Then, the phenomena associated with Raman intensity enhancement are expected to be similar in both systems. This can be corroborated by comparing the Raman spectra where it is observed that the bands at 455 and 1626 cm^−1^ are the most intense in both cases. Therefore, the adsorption and orientation of MB molecule in both SERS substrates are the same.

The EF of the Ag/PMMA composites used as SERS substrates were calculated using a similar procedure described in Section 3.2, and the SEM micrographs are shown in Figure 3b–d. The results of the EF are summarized in Table 3 and plotted in Figure 12. Contrary to the results discussed above, when Ag/PMMA composites were used as SERS substrates, the EF values obtained were one order of magnitude smaller than those obtained for Ag-NP SERS substrates. Therefore, when PMMA microspheres are present in the composite, they enhance of the Raman band intensities and, consequently, diminish the EF of the Ag/PMMA SERS substrates.

Nevertheless, analyzing the results shown in Figure 12 and Table 3, we can conclude that all composite samples show significant increment in the Raman signal with respect to those of the standard solutions (Figure 9a). Furthermore, the EF values increase as the Ag-NP concentration increases. To explain these results, we suggest that the porosity of the PMMA microspheres plays an important role. According to the theoretical results obtained from a Ag/SiO_2_ system reported by P. De León Portilla et al. [37], they probed the degree of porosity that transcends the optical response of the SiO_2_ sphere dielectric. They observed that the inner local field decreases as the porosity increases and becomes more uniform inside the dielectric sphere. To validate this, we present the textural results (see Figure 8) in which we note that PMMA microspheres are porous. Consequently, the composite samples do not exhibit higher SERS performance than the Ag-NP SERS substrates

In the same report, they studied the effect of an arrangement of dielectric and metallic spheres to resemble the uppermost surface of the artificial opal loaded with plasmonic NPs, an array like the one reported here. In their results, the SERS EF of the composites is smaller than that of the isolated metal cluster. They attributed this result to the screening effect of the electric field originating from the dielectric spheres [37]. The same effect seems to apply to our synthesized Ag/PMMA SERS substrates.

On the other hand, Fränzl et al. [38] have reported that when the stop band of a photonic crystal is coupled with the LSPR of metallic NPs, an increase in the EF of the composite used as SERS substrate is clearly observed. In this study, even when the resonance frequency of the Ag-NPs is only close to the frequency range of the stop band of the PMMA opal (see Figure 7e), an increase in the EF of the Raman bands associated with the MB adsorbed on the surface of the composite substrate is observed. Therefore, the results are expected to be considerably improved by controlling these variables on the Ag/PMMA SERS substrate, offering a promising platform for enhancing detection applications.

## 4. Conclusions

We have synthesized a set of SERS substrates using Ag-NPs and Ag/PMMA composites. In both sets, Ag-NPs were varied at three different concentrations. Ag-NPs present a quasi-spherical shape with an average diameter of 15 nm. In the case of the composites, Ag-NPs were deposited on a periodic array of PMMA microspheres (298 nm) to form an opal. In Ag/PMMA composites opals, the periodic ordering is affected by the presence of Ag-NPs. This effect was also visualized in their variable angle specular reflectance spectra in which the PBGs maxima shift towards longer wavelengths, decrease in intensity, and broaden as the concentration of Ag-NPs increases.

The SERS performance of the synthesized samples was analyzed using the detection of MB solutions with concentrations in the order of 10^−6^ M. It was found in Ag-NP SERS substrates that the EF values increase as the Ag-NP concentration increases. These results are attributed to the generation of metallic agglomerates on the film surface and, consequently, to the increment of hot spots.

We infer that the diminishing of the SERS EF of Ag/PMMA composites is probably due to the porosity of PMMA microspheres, which diminishes the local electrical field intensity, and to a screening effect caused by PMMA microspheres that affects the optical efficiency of Ag-NPs. 

Furthermore, it is expected that the EF of SERS composite substrates could be considerably improved by coupling the LSPR frequency range of Ag-NPs with the frequency range of the stop band of PMMA microsphere photonic crystal. Based on these study results, we can suggest that the synthesized SERS substrates could be applied in sensor devices for detecting and measuring different organic compounds whose low concentrations would not be easily detected using other analytic techniques.

## Figures and Tables

**Figure 1 polymers-15-02624-f001:**
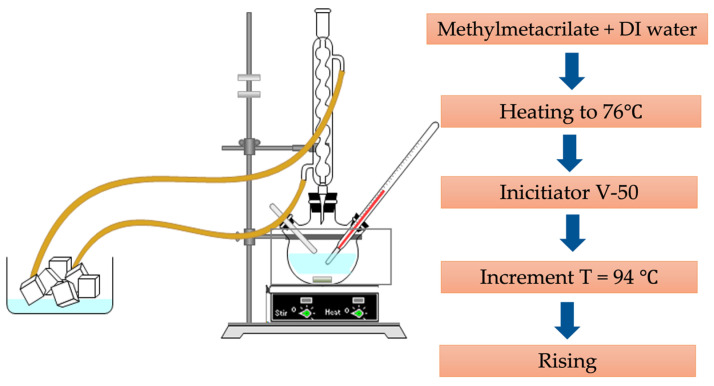
Experimental setup for the synthesis of PMMA microspheres.

**Figure 2 polymers-15-02624-f002:**
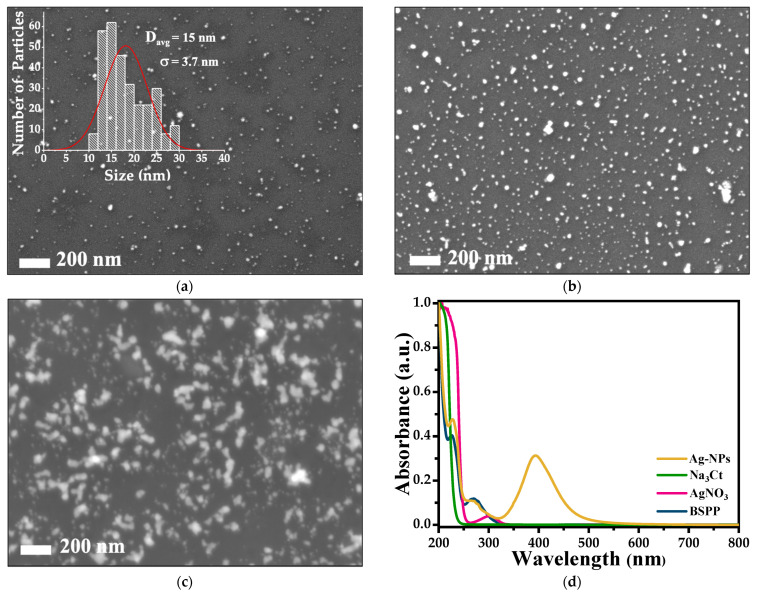
SEM micrographs, with the same scale bar, of Ag-NPs deposited on Si substrates using Ag-NP solutions of (**a**) 0.94 mM, (**b**) 1.41, and (**c**) 1.88 mM concentrations. In (**d**), UV–Vis spectra of Ag-NP solution, BSPP, NaBH_4,_ and Na_3_Ct are included for reference.

**Figure 3 polymers-15-02624-f003:**
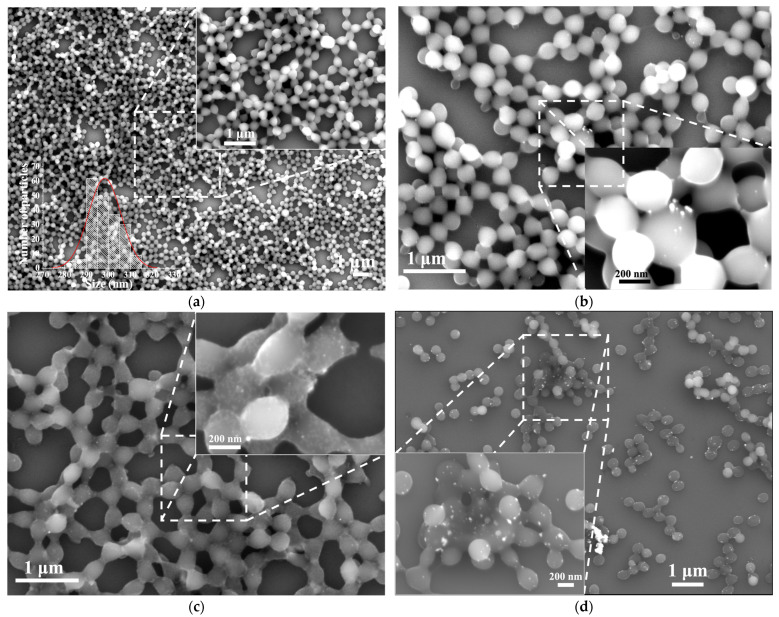
SEM micrographs, with 1 µm scale bar, of the self-assembled (**a**) PMMA microspheres, (**b**) Ag/PMMA-S1composite, and (**c**) Ag/PMMA-S2, and (**d**) corresponds to the Ag/PMMA-S3 composite. In (**a**), the PMMA microspheres size distribution histogram is included. In the insets of (**a**–**d**), images at ×100,000 are included to better visualize the Ag-NPs (white points) adsorbed on the PMMA microsphere surface.

**Figure 4 polymers-15-02624-f004:**
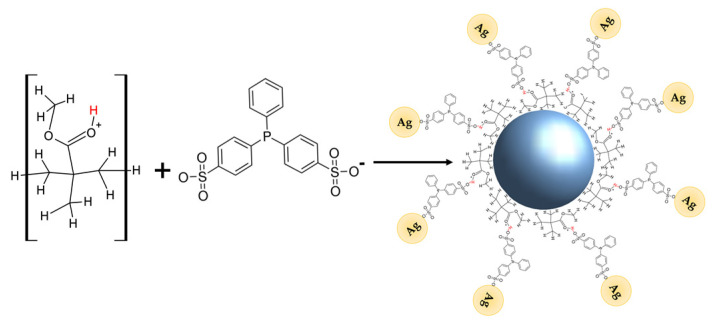
Schematic representation of hydrogen bridge type interaction forms on the surface between the sulfonate group of the Ag-NPs and the hydrogen of the protonated carbonyl group of the PMMA microspheres.

**Figure 5 polymers-15-02624-f005:**
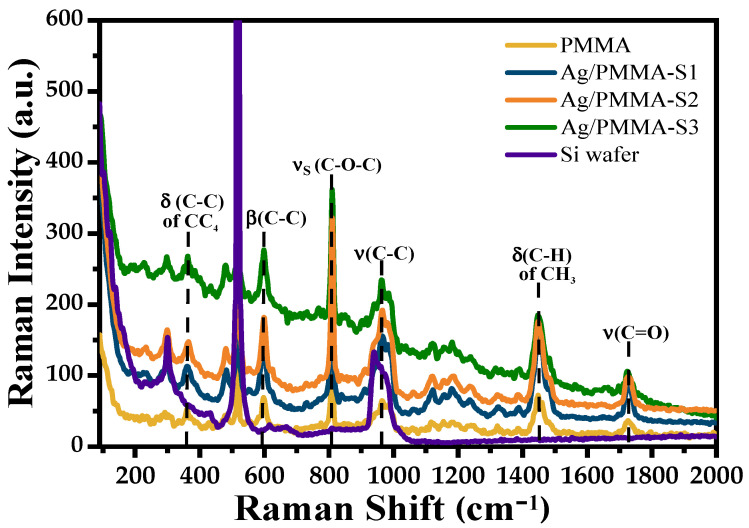
Micro-Raman spectra of PMMA and Ag/PMMA composites (Si wafer spectrum is included for reference).

**Figure 6 polymers-15-02624-f006:**
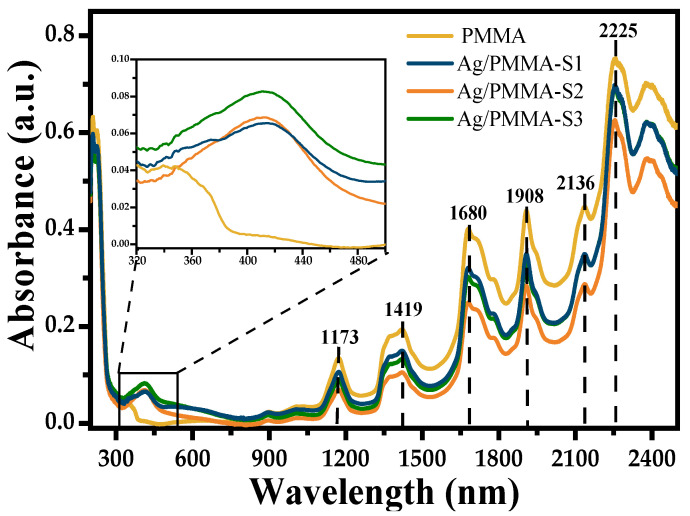
UV–Vis–NIR spectra of PMMA and Ag/PMMA composites.

**Figure 7 polymers-15-02624-f007:**
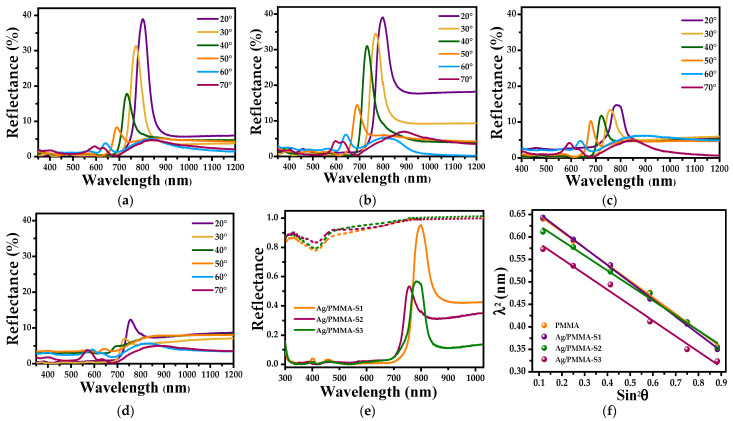
Specular reflectance spectra versus light incidence angle (θ) varied in the range of 20 to 70°, with 10° increment (Δ = 10°) of (**a**) PMMA opals and of the composites, (**b**) Ag/PMMA-S1, (**c**) Ag/PMMA-S2, (**d**) Ag/PMMA-S3, (**e**) overlapped specular reflectance spectra of all samples recorded at 20°, and (**f**) a plot of square maximum reflectance wavelength (λ^2^) versus (sin^2^θ) of PMMA opals and of Ag/PMMA composites.

**Figure 8 polymers-15-02624-f008:**
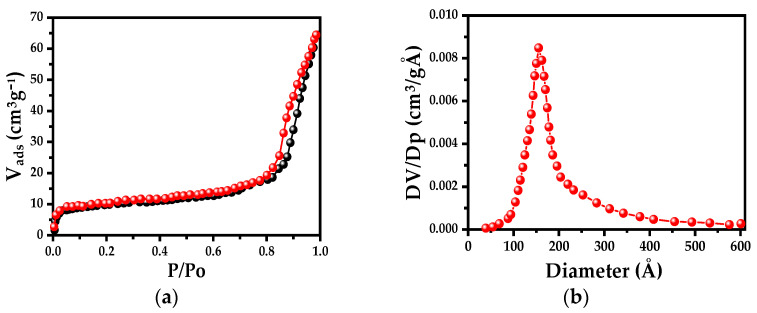
(**a**) N_2_ adsorption–desorption isotherms and (**b**) pore size distribution using the BJH method of PMMA microspheres.

**Figure 9 polymers-15-02624-f009:**
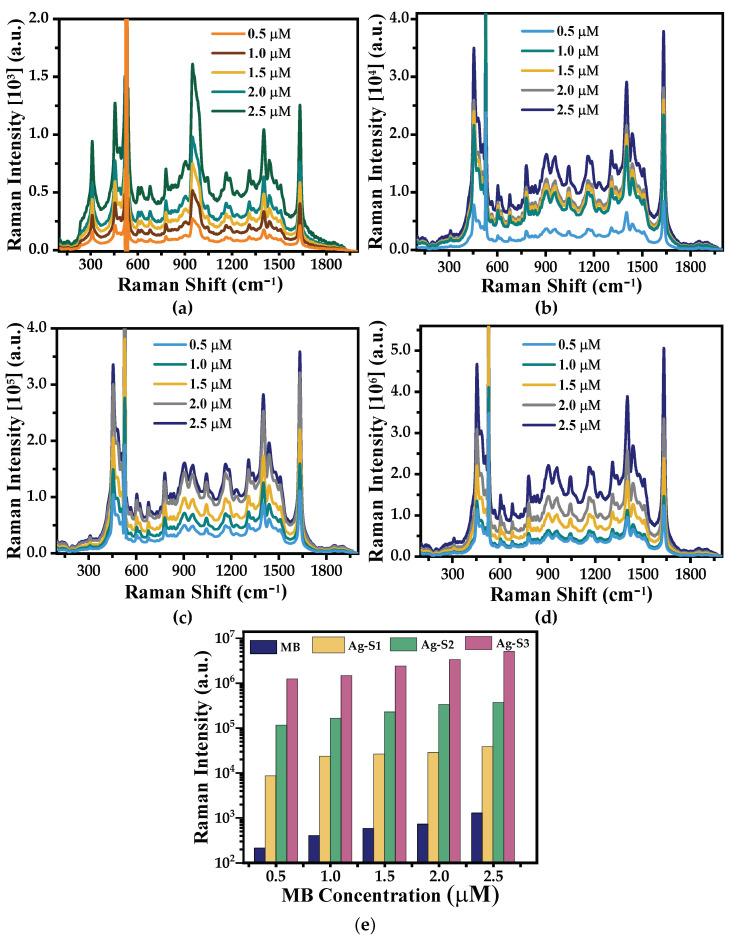
Overlapped Raman spectra recorded with (**a**) methylene blue (MB) standard solution adsorbed on Si wafer and different MB solutions (0.5 to 2.5 μM), adsorbed on (**b**) Ag-S1, (**c**) Ag-S2, and (**d**) Ag-S3, substrates, and (**e**) a bar plot of the peak intensity of the Raman band at 1632 cm^−1^ from all SERS substrates including the one of MB for comparison.

**Figure 10 polymers-15-02624-f010:**
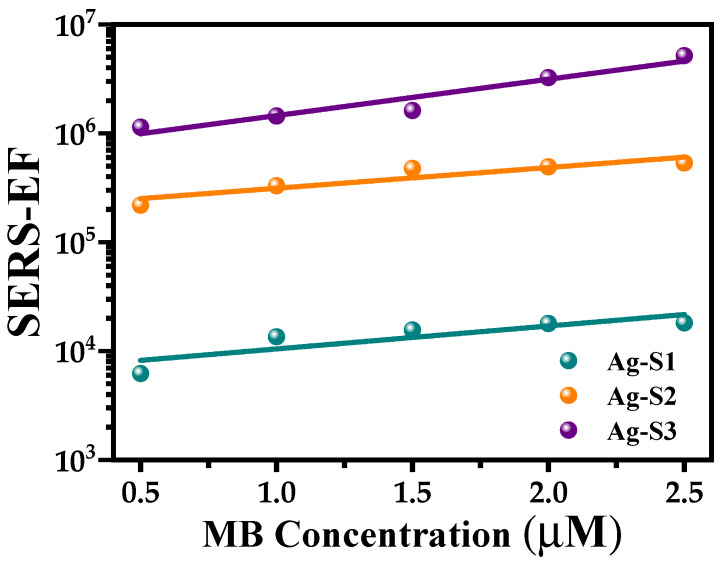
A plot of the enhancement factor (EF) data of the band at 1632 cm^−1^ of the Raman spectra of SERS substrates: Ag-S1, Ag-S2, and Ag-S3 versus methylene blue (MB) concentration.

**Figure 11 polymers-15-02624-f011:**
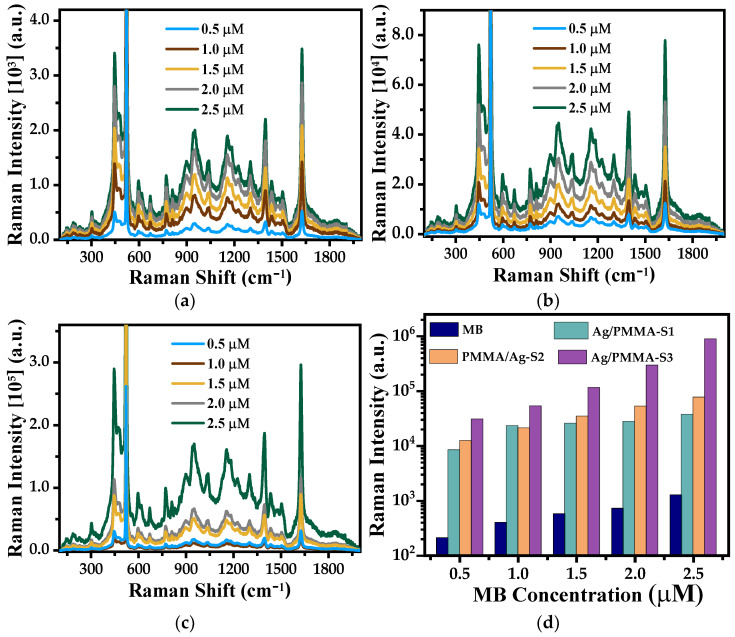
Overlapped Raman spectra recorded from SERS substrates with adsorbed aqueous solutions of MB (0.5 to 2.5 μM): (**a**) Ag/PMMA-S1, (**b**) Ag/PMMA-S2, and (**c**) Ag/PMMA-S3, respectively, and (**d**) selected peak intensity of the Raman band at 1632 cm^−1^ from all SERS substrates including the one of methylene blue (MB) for comparison.

**Figure 12 polymers-15-02624-f012:**
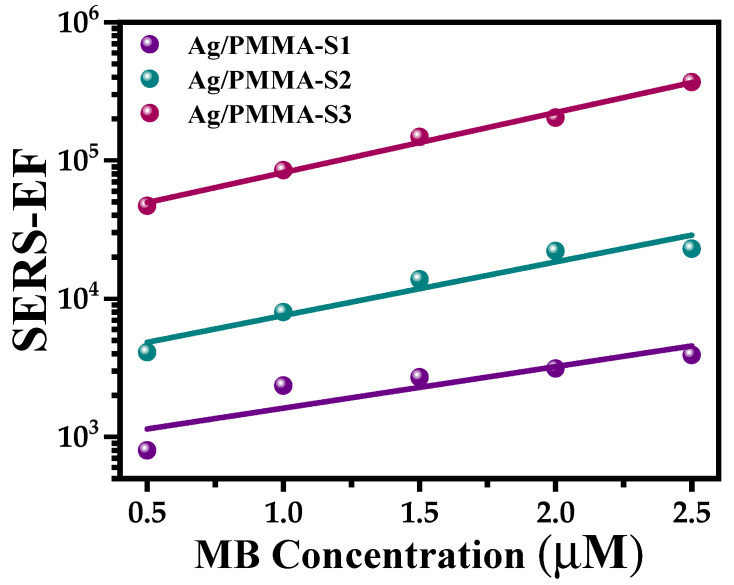
A plot of the enhancement factor (EF) data of the band at 1626 cm^−1^ of the Raman spectra recorded from all SERS substrates, Ag/PMMA-S1, Ag/PMMA-S2, and Ag/PMMA-S3 versus methylene blue (MB) concentration.

**Table 1 polymers-15-02624-t001:** Wavenumbers (cm^−1^) and assignment Raman bands of PMMA and Ag/PMMA composites [23].

PMMA	Ag/PMMA	Assignment
365	359	δ(C-C) of CH_4_
601	597	β(C-C)
813	810	$\nu_{s}$(C-O-C)
969	965	ν(C-C)
990	985	ν(C-C)
1066	1060	ν(C-C)
1123	1120	ν(C-O), ρ(CH_3_)
1159	1155	$\nu_{a}$(C-O-C)
1185	1180	$\nu_{a}$(C-O-C)
1240	1235	ν(C-O)
1326	1320	τ(CH_2_)
1390	1386	δ(C-H) of CH_3_
1452	1447	δ(C-H) of CH_3_
1481	1475	δ(C-H) of CH_2_
1728	1725	δ(C=O)

δ—in-plane deformation, β—in-plane bending, ν—stretching vibration, s—symmetrical, a—asymmetrical, ρ—rocking vibration, τ—twisting.

**Table 2 polymers-15-02624-t002:** MB concentrations and EF data for the different Ag-NP SERS substrates.

MB Concentration	SERS EF		
(×10^6^ M)	Ag-S1 (×10^4^)	Ag-S2 (×10^5^)	Ag-S3 (×10^6^)
0.5	0.62 ± 0.03	2.18 ± 0.07	1.14 ± 0.12
1.0	1.35 ± 0.04	3.30 ± 0.08	1.44 ± 0.13
1.5	1.56 ± 0.05	4.75 ± 0.07	1.62 ± 0.15
2.0	1.79 ± 0.04	4.92 ± 0.08	3.25 ± 0.11
2.5	1.81 ± 0.05	5.34 ± 0.09	5.18 ± 0.09

EF: Enhancement factor; MB: methylene blue.

**Table 3 polymers-15-02624-t003:** MB concentrations and EF data for the different Ag/PMMA SERS substrates.

MB Concentration	SERS EF		
(×10^6^ M)	Ag/PMMA-S1 (×10^3^)	Ag/PMMA-S2 (×10^4^)	Ag/PMMA-S3 (×10^5^)
0.5	0.80 ± 0.11	0.41 ± 0.05	0.47 ± 0.08
1.0	2.35 ± 0.07	0.80 ± 0.07	0.85 ± 0.09
1.5	2.70 ± 0.09	1.38 ± 0.09	1.48 ± 0.12
2.0	3.12 ± 0.14	2.21 ± 0.09	2.04 ± 0.10
2.5	3.91± 0.12	2.30 ± 0.08	3.69 ± 0.11

EF: Enhancement factor; MB: methylene blue.

## Data Availability

The data presented in this study are available on request from the corresponding author.

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
