# Peer review of "Surface-Enhanced Raman Scattering (SERS) Substrates Based on Ag-Nanoparticles and Ag-Nanoparticles/Poly (methyl methacrylate) Composites"

_polymers, 2023, doi:10.3390/polym15122624_

Round 1
Reviewer 1 Report
1- Re-write the abstract to be concise and informative and include the key numerical results obtained
2- Add a paragraph at the end of the introduction to show the author's motivation and the novelty of this work relative to the previously published papers.
3- In the introduction please cited the recent literature related to the SERS
For example
a. https://www.nature.com/articles/s43586-021-00083-6
b. https://pubmed.ncbi.nlm.nih.gov/31478375/
c. https://www.nature.com/articles/srep25307
d. https://pubs.acs.org/doi/10.1021/jacs.2c09050
e. https://pubs.rsc.org/en/content/articlelanding/2022/an/d1an02165f
f. https://www.mdpi.com/2079-4991/12/21/3778/htm
4- In Figure 4f change the X-axis to sin2ÆŸ instead of sen2ÆŸ
5- As you ascribed the enhancement to the porosity of the prepared samples and chemical enhancement, so please provide PET analysis to identify the porosity parameters and surface area
6- Provide the obtained key results in the conclusion.
7- The language needs more revision
Reviewer 2 Report
The manuscript titled “Surface-Enhanced Raman Scattering (SERS) substrates based on Ag-nanoparticles and Ag-nanoparticles/Poly (methyl metacrylate) Composites” is about synthesis and the evaluation of Ag-NPs and Ag/PMMA composites used as SERS substrates to improve the Raman scattering signal of methylene blue (MB) used as a probe molecule with concentrations as low as 5x10-7 M. The topic is interesting, the experiment is well-designed and, in general, the work is worthy of consideration. However, in present form the manuscript needs to be revised. Some comments and recommendations are listed below:
In paragraph 4 of Introduction, the authors state that Ag NPs have advantages in comparison with other noble metal NPs. However, the next two paragraphs are about composites with metal NPs (Ag NPs are not specified). Logically, the paragraph 4 should be moved below paragraphs 5,6 to point out that among all metals used in considering composites the Ag NPs are one of the most promising. Or inverse, paragraph 4 can be left on the same place, but paragraphs 5,6 should be rewritten considering complexes with Ag NPs.
Paragraphs 4,5,6. Supporting references are extremely needed.
Why were Ag nanoparticles chosen as a nanomaterial? Why not other precious metals? Or why not bimetallic or polymetallic particles? The authors need to describe this in the field of relevance of the use of the material.
Consequently, paragraph 4 should be expended well. The recent studies on Ag NPs synthesis, characterization and application should be discussed to point out their advantages, importance, relevance. Some recent studies can be considered: https://doi.org/10.3390/mi13071105, https://doi.org/10.1155/2021/6687290, https://doi.org/10.1186/s11671-019-3117-5
In section 2.2, the authors indicated the synthesis of microspheres. But the section needs to be supplemented with a scheme for obtaining the material, with a details of the synthesis parameters.
For all equipment the next information should be given: brand, manufacturer and localization (city, country)
In Results and Discussion, the authors should present the dependence of the position and intensity of the absorption maximum on the concentration of Ag. This function will reliably determine the optimal concentration of Ag.
In Raman spectra, the authors should provide a decoding of the main bands – to present the modes of oscillations, in the form of a table or in text format.
For a comprehensive presentation of the results of the study, the authors should present the dependence of the average hydrodynamic radius of the particles on the concentration, supported by a schematic representation of the distribution of nanoparticles on the material.
More recent references should be used In discussion to support the results and statements or hypothesis.
All abbreviations in figures and tables should be defined under the figure/table, even if it was defined in the text before.
Ag-NPs or Ag NPs?
L. 243. Why the authors decided to introduce abbreviations one more time?
Conclusions can be reached by more important data obtained. Future prospects should be expanded and explained better. Why the authors decided that the synthesized SERS substrates can be applied in sensor devices for detecting and measuring different organic compounds whose low 436 concentrations could not be easily detected with other analytic techniques. If I am right it was not discussed in the main text and it is not mentioned in aim or objectives.
42% of cited references were published more than 5 years ago. The authors should use more recent works in References and replace some old sources where is possible.
Round 2
Reviewer 1 Report
Authors respond to some of my previous comments, however, there are some issues that need more consideration;
1- The text is not sufficiently clear and the use of English should be improved.
For example :
Line 17 : "composites" should be " composites,"
Line 18: "altereded " should be " altered"
2- The abstract still lacks a clear statement of the research question or hypothesis, which makes it difficult to evaluate the relevance and contribution of the study. Also, the abstract lacks a clear statement of the implications or potential impact of the research on the field o SERS
3- Still the novelty statement of this study is unclear
Reviewer 2 Report
The authors considered all comment in revision. The revised manuscript can be recommended for publication
Author Response
The introduction and the conclusion have been improved. The text and the English redaction have been improved.